# SPEECH-MLP: A SIMPLE MLP ARCHITECTURE FOR SPEECH PROCESSING

## ABSTRACT

Transformers have shown outstanding performance in recent years, achieving state-of-the-art results in speech processing tasks such as speech recognition, speech synthesis and speech enhancement. In this paper, we show that, despite their success, such complex models are not needed for some important speech related tasks, which can be solved with much simpler and compact models. Thus, we propose a multi-layer perceptron (MLP) architecture, namely speech-MLP, useful for extracting information from speech signals. The model splits feature channels into non-overlapped chunks and processes each chunk individually. These chunks are then merged together and further processed to consolidate the output. By setting different numbers of chunks and focusing on different contextual window sizes, speech-MLP learns multiscale local temporal dependency. The proposed model is successfully evaluated on two tasks: keyword spotting and speech enhancement. In our experiments, two benchmark datasets are adopted for keyword spotting (Google speech command V2-35 and LibriWords) and one dataset (VoiceBank) for speech enhancement. In all experiments, speech-MLP surpassed the transformer-based solutions, achieving better performance with fewer parameters lower GFLOPS. Such results indicate that more complex models, such as transformers, are oftentimes not necessary for speech processing tasks. Hence, simpler and more compact models should always be considered as an alternative, specially in resource-constrained scenarios.

## 1 INTRODUCTION

As in many machine learning disciplines, speech processing is embracing more and more complex models, where transformer (Vaswani et al., 2017) is a particular example. It was first proposed to tackle machine translation, and afterwards was successfully applied to multiple research fields such as natural language processing (NLP) (Devlin et al., 2018) and computer vision (CV) (Dosovitskiy et al., 2020). The core of the transformer model is a self-attention mechanism, by which any two elements in a sequence can interact with each other, hence capturing long-range dependency. Considering that speech signals are naturally temporal-dependent, researchers in the speech community recently explored transformer-based models in multiple speech processing tasks, and remarkable performance was reported in speech recognition (Dong et al., 2018; Karita et al., 2019; Huang et al., 2020), speech enhancement (SE) (Kim et al., 2020; Fu et al., 2020), keyword spotting (KWS) (Berg et al., 2021; Vygon & Mikhaylovskiy, 2021) and speech synthesis (Li et al., 2019). Recently, the conformer architecture, which combines convolution and self-attention, achieved excellent success in speech processing tasks and attracted much attention Gulati et al. (2020).

In this paper, we ask the following question: Do we need complex models such as transformers for certain speech processing tasks?

This question is closely related to the principle of 'parsimony of explanations', a.k.a., Occam's razor (Walsh, 1979). According to this principle, if there is any possibility, we should seek the models that can represent the data with the least complexity (Rasmussen & Ghahramani, 2001; Blumer et al., 1987). However, in the public benchmark tests, complex and elaborately designed models are often ranked higher, due to the better *reported* performance. For example, the KWS benchmark on Google

speech command[1] and the SE benchmark on VoiceBank+DEMAND[2], transformer-based models are among the top ranks. Although the good performance is celebrating, the increased model complexity implies potential over-tuning and over-explanation, the risk that the Occam's razor principle intends to avoid.

We, therefore, attempt to discover the *simplest* neural architecture, that is powerful enough to achieve comparable performance as the best existing models, in particular transformers, while eliminating unnecessary complexity. Our design is based on domain knowledge, in particular, three properties of speech signals: (1) **temporal invariance**, (2) **frequency asymmetry**, and (3) **short-term dependency** (Huang et al., 2001; Benesty et al., 2008; Furui, 2018). Based on these knowledge, we build the speech-MLP, a simple multi-layer perceptron (MLP) architecture, shown in Fig. 1. Besides the normalization components, the architecture involves simple linear transformations only. The core of the architecture is the Split & Glue layer, which splits the channel dimension into multiple chunks, processes each chunk separately, and finally merges the processed chunks in order to attain the output. Speech-MLP processes each time frames independently (compatible to temporal invariance), and the splitting & gluing procedure allows different treatments for different frequency bands (compatible to frequency asymmetry), and involves the local context of multiple scales (compatible to short-term dependency),

We tested the model on two speech processing tasks: keyword spotting with the Google speech command V2-35 and Libriword benchmark datasets; and speech enhancement with the VoiceBank benchmark dataset. Results showed that on both tasks the proposed speech-MLP outperforms complex models, in particular models based on transformers. Such results demonstrate that by utilizing domain knowledge and employing appropriate normalization techniques, it is possible to design simple yet powerful models. In some cases, these simple models even beat complex models on open benchmarks, where complex models are more likely to obtain good performance by careful tuning.

In summary, we proposed Speech-MLP, a simple yet effective neural model to represent speech signal. On the KWS and SE tasks, we demonstrated that the simple model can achieve performance comparable to or even better than transformers with less parameters and inference time. Our work shows that by taking domain-knowledge into account, it is possible to remove unnecessary complexity (e.g., modeling for the long-range dependency in KWS and SE) in model design, as advocated by the Occam's razor.

## 2 RELATED WORK

Recent research has shown that a simple model can be as effective as complex and task specific models such as transformers in some important tasks. In (Tolstikhin et al., 2021), for example, the authors proposed a simple architecture for vision, namely MLP-Mixer. The model receives a sequence of image patches and performs channel-wise and patch-wise linear projection alternatively and iteratively. Without using convolutions or self-attention, the Mixer architecture separates the per-location (channel-mixing) and cross-location (token-mixing) operations (Tolstikhin et al., 2021). While the channel-mixing MLPs enable communication between different channels, the token-mixing MLPs allow communication between different spatial locations (tokens). Tested on image classification benchmarks, MLP-Mixer achieved performance comparable to SOTA models, in particular the vision transformer model (Tolstikhin et al., 2021).

In another recent work (Liu et al., 2021), the authors investigated the need of the self-attention mechanism in transformers, proposing an alternative MLP-based architecture, namely gMLP. The model, based on MLP layers with gating, consists of a stack of L identical blocks. Each block comprises a normalization layer, a channel projection, followed by an activation function and a spatial gating unit, followed by another channel projection (Liu et al., 2021). It achieves similar performance when compared to the vision transformer model (Touvron et al., 2021b), being 3 % more accurate than the aforementioned MLP-mixter model with 66 % fewer parameters. The model was also successful on language modeling in the BERT setup (Liu et al., 2021), minimizing perplexity as well as Transformers. The authors also found that perplexity reduction was more influenced by the model capacity than by the attention mechanism.

---

[1] https://paperswithcode.com/sota/keyword-spotting-on-google-speech-commands
[2] https://paperswithcode.com/sota/speech-enhancement-on-demand

Inspired by vision transformers (Touvron et al., 2021b)(Dosovitskiy et al., 2020), in (Touvron et al., 2021a), the authors apply the skip connection technique from ResNet's to MLP layers and propose the so-called Residual Multi-Layer Perceptrons (ResMLP). The model receives non-overlapping image patches, typically $16 \times 16$. These patches go through a linear transformation in order to attain d-dimensional embeddings. The embeddings are then fed to a sequence of ResMLP blocks to produce a set of d-dimensional output embeddings. An average pooling is applied on the d-dimension output vector to represent the image, a linear classifier is used then to predict the label associated with the image (Touvron et al., 2021a).

Differently from Mixer-MLP, gMLP and ResMLP, CycleMLP can process inputs of arbitrary resolution with linear computational complexity as its receptive fields are enlarged for context aggregation (Chen et al., 2021). The model is based on Cycle Fully-Connected Layer (Cycle FC), serving as a generic, plug-and-play transformer-free architecture. Results show CycleMLP outperforming existing MLP-like models on ImageNet classification, achieving good performance on object detection, instance segmentation and semantic segmentation (Chen et al., 2021).

The aforementioned research highlights that, despite their success, convolution and self-attention mechanisms are not mandatory for some CV and NLP tasks, and can be replaced by simpler layers such as MLP with a customized design. Although typical convolution operations are not used by these MLP solutions (but rather $1 \times 1$ convolution as pointed out in (Chen et al., 2021) and (Tolstikhin et al., 2021)), these MLP approaches are inspired by CNN architectures for computer vision related tasks. Their building block, nonetheless, is similar and based on applying linear transformation on spatial locations and feature channels.

Although inspired by these new MLP architectures, speech-MLP focuses on speech signals rather than images. This implies in processing different input resolutions given the nature of the input signal. The split & glue layer is very similar to a separable CNN (Chen et al., 2018), if we regard the frame-independent processing as 1-D convolution in time. In particular, it is essentially a group-wised CNN (Romero et al., 2020) with different kernels for each group. However, from the perspective of feature learning, the entire split & glue is an MLP if our focus is a particular frame (within a context). That is why a 1-D convolution is often called a time-delay neural net (TDNN) (Waibel et al., 1989). We follow this convention and name our structure as speech-MLP.

A key motivation of the speech-MLP structure is to respect the properties of speech signals. It should be emphasized that almost all successful techniques in speech processing take these properties into account, for instance the hidden Markov model (HMM) assumes short-term dependency (Rabiner & Juang, 1986), TDNN assumes temporal invariance (Waibel et al., 1989), and frequency asymmetry is explicitly implemented in the famous MFCC feature (Mermelstein, 1976). In this paper, the role of knowledge of speech signals is to help remove unnecessary complexity, i.e., seeking the minimum structure that make reflect these basic properties.

Finally, MLP is not new in speech processing; in fact the neural models used in early days in speech processing are all general MLPs, e.g., (Bourlard & Morgan, 2012). Speech-MLP is a special designed MLP, by taking the properties of speech signals into account.

## 3 METHODOLOGY

Our model, referred to as speech-MLP, is presented in Figure 1. Note that for a given speech waveform, a sequence of acoustic features, denoted by $X = \{x_1, x_2, ..., x_n\}$, are first extracted. These features are then fed into $N$ stacked speech-MLP blocks and the output of the last speech-MLP block is a speech representation that needs to undergo task-specific layers in order to perform specific tasks, such as the ones addressed in this study: SE and KWS.

Inside of each speech-MLP block, there are three components: (1) a linear transformation for a pre-projection of the extracted acoustic features; (2) a Split & Glue layer for processing the projected acoustic features while addressing frequency asymmetry and temporal dependency, and (3) another linear transformation for post-projection of the final representation. Two residual connections are also adopted to encourage gradient propagation. The first one maps the input features onto the output of the last linear transformation (i.e., the output of the post-projection operation). The second residual connection maps the output of the first linear transformation (i.e., the output of the pre-projection operation) onto the output of the Split & Glue layer. Note that normalization tech-

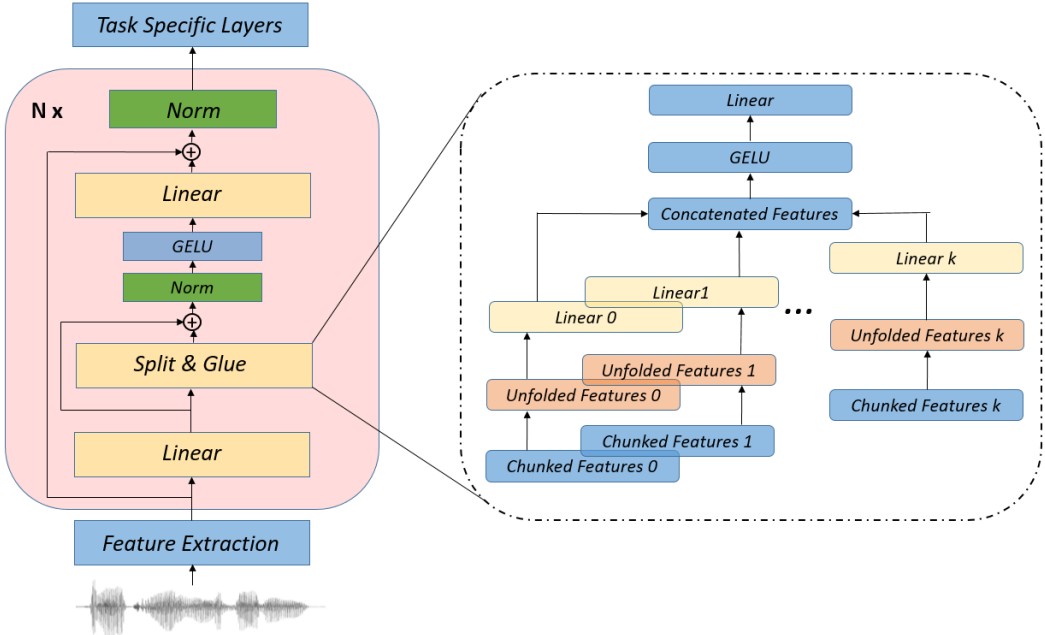

Figure 1: Proposed architecture consisting of $N$ speech-MLP blocks, comprising three components: (1) a linear transformation as pre-projection; (2) a Split & Glue layer as the main processing module; and (3) another linear transformation as post-processing.

niques are also applied to regulate the feature distribution (by layer norm) and temporal variance (by instance norm). In the next section, we give more details on the Split & Glue layer, followed by a discussion on the normalization methods adopted in this work.

## 3.1 SPLIT & GLUE

Figure 2 depicts how the Split & Glue layer operates. The sequence of acoustic features is denoted by $X \in \mathcal{R}^{H \times T}$, with $T$ and $H$ being, respectively, the length and the number of channels of the input sequence. The first step is to split $X$ into $K$ non-overlapping *chunks*, as illustrated in both Figure 1 and Figure 2. The split referred to as $X \to \{X^1, .., X^k, .., X^K\}$, is performed along the channel dimension. In our experiments, the channel dimension of each chunk is considered the same, leading to $X^k \in \mathcal{R}^{H/K \times T}$. For each chunk, $X^k$, a context expansion is then performed through the so-called unfolding operations. This results in context-expanded chunks, denoted by $X_w^k \in \mathcal{R}^{w^k H/K \times T}$, where $w^k$ is the size of the context window induced by the unfolding operation.

Note that the number of chunks $K$ and the window size $w^k$ can be arbitrarily selected for each chunk. This flexibility allows us to represent multi-scale contexts by adopting different window sizes for different chunks. In Figure 2, for instance, the input channels are split into two chunks, and the window sizes are set to $3$ and $5$, respectively. This leads to the model learning from small and large contexts simultaneously.

The unfolded chunk $X_w^k$ is projected by a linear transformation, leading to a new representation for the initial chunk, $Y^k \in \mathcal{R}^{\hat{H} \times T}$, where $\hat{H}$ could be set arbitrary and is called the number of *Glue channels*. We highlight that the linear transformation used in the above chunk-wise operation is shared across all the time steps for a single chunk, and each time frame is processed independently. This setting reduces the number of parameters and is compatible with the temporal invariance property of speech signals. Nevertheless, different weight parameters are adopted for different chunks, to provide sufficient flexibility.

Finally, all the learned speech representations, $Y^i$, are concatenated along the channel dimension, forming a glued feature matrix $Y^G = \{Y^1, Y^2, ..., Y^K\}$. Following, another linear transformation

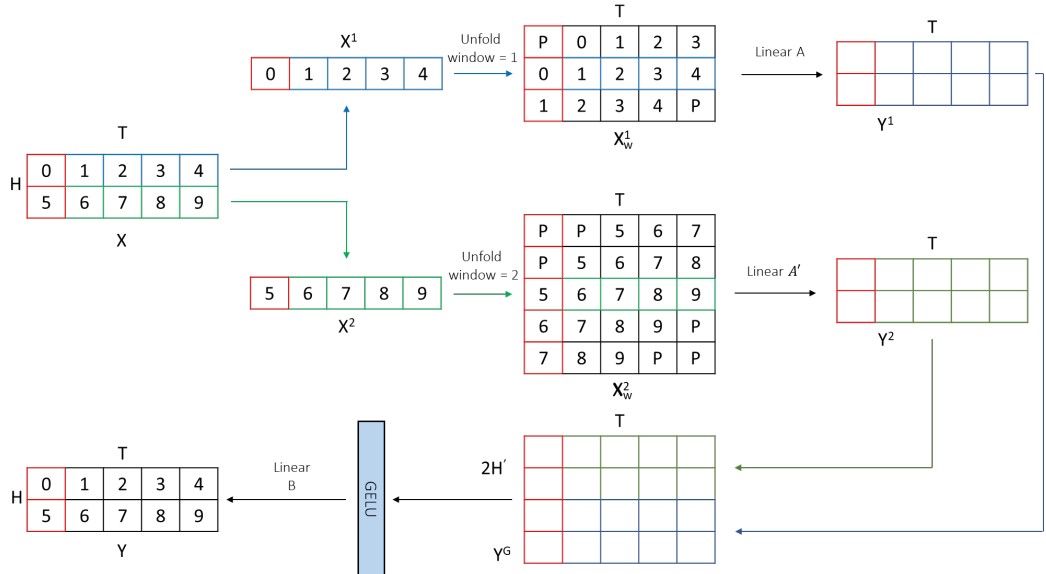

Figure 2: The Split & Glue layer, best viewed in color. The input feature X is split to 2 chunks (denoted by blue and green box respectively), and the window sizes of the unfold operation are set to 3 and 5 respectively for the two chunks. The red box indicates a single time step, and 'P' in $X_w^1$ and $X_w^2$ represents padding.

is applied in order to obtain the output feature $Y \in \mathcal{R}^{H \times T}$. Again, the linear transformation is shared across all the time steps, to reflect temporal invariance.

## 3.2 NORMALIZATIONS

Normalization plays an important role in our speech-MLP model. We employed two normalization approaches: (1) layer normalization (LN) (Ba et al., 2016) and (2) instance normalization (IN) (Ulyanov et al., 2016).

Layer normalization is applied across the channel dimension at each time step. Thus, it computes statistics (mean and variance) on each column of $X \in \mathcal{R}^{H \times T}$, and then uses these statistics to normalize the elements in the same column. With this normalization technique, the distribution of the feature vector at each time step is regularized.

Instance normalization is used to perform per-channel normalization. That is, the statistics are computed on each row of $X \in \mathcal{R}^{H \times T}$ and applied across the time steps to normalize the elements of each row. Thus, the temporal variation of each channel is normalized. Note that IN extends the conventional cepstral mean normalization (CMN) approach (Liu et al., 1993), by normalizing not only acoustic features, but also features produced by any hidden layer.

Empirically, we found that IN was only effective for the SE task while the LN was more important for the KWS task. Therefore, we apply LN only for KWS and IN for SE.

## 4 EXPERIMENTS

We evaluate the proposed speech-MLP model in two speech processing tasks: speech enhancement and keyword spotting. In this section, we introduce these tasks and their respective datasets, used in our experiments, followed by experimental settings, experimental results, and the ablation study.[3]

---

[3]The code will be available on github. To respect the double-blind review, the link will be sent to the reviewers when the discussion is open.

Table 1: Adopted architectures for the KWS and SE tasks. In the KWS setting, (S) and (L) denote small model and large model respectively.

|  |  | KWS | SE |
|---|---|---|---|
| Linear 0 | Input Channels | 40 | 257 |
|  | Output Channels | 128(S/L)/256(XL) | 256 |
|  | Bias | true | true |
| Speech-MLP | #Blocks | 4(S/L)/12(XL) | 10 |
|  | Input Channels | 128(S/L)/256(XL) | 256 |
|  | Glue Channels | 60(S)/100(L)/120(XL) | 60 |
|  | Hidden Channels | 40(S)/80(L)/100(XL) | 40 |
|  | Bias | true | true |
|  | Context Window | $\{3, 7, 9, 11\}$ | $\{3, 7, 9, 11\}$ |
|  | Normalization | Layer Norm | Instance Norm |
|  |  | - | GELU |
|  |  | MaxPooling | - |
| Linear 1 | Input Channels | 128 | 256 |
|  | Output Channels | 128 | 257 |
|  | Bias | true | true |
|  |  | GELU | - |
| Linear 2 | Input Channels | 128 | - |
|  | Output Channels | #Keywords | - |
|  | Bias | true | - |

## 4.1 KEYWORD SPOTTING

Keyword spotting aims at detecting predefined words in speech utterances (Szöke et al., 2005; Mamou et al., 2007; Wang, 2010; Mandal et al., 2014). In our experiments, we explore two KWS datasets: (1) the Google speech commands V2 dataset (Warden, 2018), and (2) the LibriWords (Vygon & Mikhaylovskiy, 2021). The Google speech commands V2 dataset (here, referred to as **V2-35**) consists of $105,829$ utterances of 35 words, recorded by 2,618 speakers. The training, validation and test sets contain $84,843$, $11,005$ and $9,981$ utterances respectively. The LibriWords dataset, larger and more complex, is derived from 1000-hours of English speech from the LibriSpeech dataset (Panayotov et al., 2015). Signal-to-word alignments were generated using the Montreal Forced Aligner (McAuliffe et al., 2017) and are available in (Lugosch et al., 2019). The averaged duration of the keywords are $0.28$ seconds. The provider defined four benchmark tests, based on the number of target keywords: **LW-10**, **LW-100**, **LW-1K** and **LW-10K**, where the target keywords are 10, 100, 1k and 10k respectively. More details on this dataset are presented in Appendix.

### 4.1.1 SETTINGS

We used the same architecture in all the KWS tasks, except that the dimension of the output layer was adapted to the number of keywords, as shown in Table 1. Note that we set the window size $w$ to be $\{3, 7, 9, 11\}$. This allows us to exploit multi-scale contexts. Additionally, we set the stride to be 1 and appropriately set the padding list $p$ to ensure that all the expanded features are in the same length and equal to that of the input feature.

Prior to the feature extraction step, each speech recording is resampled to 16 kHz. Then, 40-dimensional Mel-Frequency Cepstral Coefficients (MFCC) are attained as the acoustic features. The MFCC features are then projected target dimensional feature vector by a linear layer and then forwarded to speech-MLP blocks. The output features are then passed through a max-pooling operation collects the information across time steps. Finally, two linear layers with a GELU activation function in the middle and a softmax activation are employed in order to attain the posterior probabilities that the input speech belongs to each keyword. For regularization we used SpecAugment (Park et al., 2019), dropout (Baldi & Sadowski, 2013), and label smoothing (Müller et al., 2019) were used to prevent overfitting.

Three model architectures have been verified in all the experiments: a 180k small model denoted by **Speech-MLP-S**, a 480k large model denoted by **Speech-MLP-L**, and a 2375K extra large model

Table 2: Performance comparison on KWS tasks in terms of accuracy (%).*The size is reported from the entire KWS model on the V2-35 task. Due to the different numbers of keywords, the value varies from task to task, although the backbone is the same. #The Gflops is calculated on V2-35 task.

| Models | GFlops# | Size* | Accuracy% | | | | |
|---|---|---|---|---|---|---|---|
| | | | V2-35 | LW-10 | LW-100 | LW-1K | LW-10K |
| Att-RNN (de Andrade et al., 2018) | - | 202k | 93.90 | - | - | - | - |
| KWT-1 (Berg et al., 2021) | 0.108 | 607k | $96.85 \pm 0.07$ | - | - | - | - |
| KWT-2 (Berg et al., 2021) | 0.469 | 2394k | $97.53 \pm 0.07$ | - | - | - | - |
| KWT-3 (Berg et al., 2021) | 1.053 | 5361k | $97.51 \pm 0.14$ | - | - | - | - |
| Res15-CE (Vygon & Mikhaylovskiy, 2021) | - | 237k | 95.96 | 88.8 | 82.3 | 78.2 | 69.3 |
| Res15-TL (Vygon & Mikhaylovskiy, 2021) | - | 237k | 97.00 | 91.7 | 86.9 | 84.3 | 81.2 |
| Speech-MLP-S (Ours) | 0.016 | 180k | $97.15 \pm 0.07$ | 95.03 | 90.91 | 90.16 | 89.16 |
| Speech-MLP-L (Ours) | 0.046 | 480k | $97.36 \pm 0.16$ | 95.37 | 92.11 | 91.50 | 90.82 |
| Speech-MLP-XL (Ours) | 0.228 | 2375K | $\mathbf{97.56} \pm 0.09$ | **95.80** | **93.22** | **93.27** | **93.01** |

denoted by **Speech-MLP-XL**. The three models are different in the number of channels of the hidden layer (i.e., after the pre-projection) and the channels within the Split & Glue block (i.e., channels after Linear A, and layers in Fig. 2), as shown in Table 1.

For the experiments on the Google speech commands dataset, we applied the following data augmentation techniques: time shifting, audio re-sampling and noise perturbation: as in (Berg et al., 2021; Vygon & Mikhaylovskiy, 2021). After augmentation, the data was increased to 10 times the size of **V2-35**. We set the batch size to be 256 and trained the model for 100 epochs on 4 cards V100 Nvidia GPU.

For the experiments on the LibriWords, the batch size was set to 1024, and we trained the model for 20 epochs on 2 cards V100 Nvidia GPU which showed to be enough for this dataset. The training schemes were set differently simply because Libriwords is huge and long-term training is not economic.

The performance of the proposed model is compared to three benchmarks. The first one referred to as Att-RNN, is a CNN-LSTM architecture with the attention mechanism introduced in (de Andrade et al., 2018). The model has approximately 202k trainable parameters and attains reasonable performance. Another recent solution, based on a transformer architecture is adopted as the second benchmark (Berg et al., 2021). We refer to this benchmark as KWT-$K$ where $K$ refers to different size of models. Res15 (Vygon & Mikhaylovskiy, 2021), another recent work based on ResNet reports high performance on both V2-35 and Libriwords. The authors reported results with two configurations, one trained by cross entropy (Res15-CE) and the other based on triple loss (Res15-TL). We use them as the third benchmark.

### 4.1.2 RESULTS

Table 2 presents the results of the benchmarks discussed in the previous section and the performance of the proposed Speech-MLP, the experimental results are presented by mean value and 95% confidence of 5 trials with different random seeds on V2-35. It can be observed that the Speech-MLP models outperform all the benchmarks with comparable model sizes. Note that the small version of speech-MLP, which contains less than half of the parameters of its large version, can still maintain reasonable performance, providing higher accuracy than most benchmarks. The performance of our solution on the Libriword dataset is even more significant. It outperforms Res15-CE and Res15-TL while being able to maintain performance across all LibriWord dataset sizes. Our conjecture is that by the knowledge-driven design, we can use the parameters more efficiently, which allows for the use of smaller models to handle large-scale tasks.

### 4.1.3 ABLATION STUDY

To investigate how each module impacts the performance of speech-MLP, we conducted an ablation study, in order to fair compare each model we use fixed random seed 123 in all ablation study experiments, we show that window list to {3} equivalent to use TDNN with kernel size to 3, and window list to {3, 3, 3, 3} equivalent the TDNN with 4 groups convolution operation with kernel size

to 3 in split & glue layer, and our proposed speech-MLP with a variance of window sizes outperform these existing solutions. We particularly focus on the chunk splitting, specially the number of chunks and the context window of each chunk. They are the only hyperparameters that we need to design in speech-MLP, by using domain knowledge.

The results are reported in Table 3. It can be observed that the setting for the number of chunks and the context window does matter. A longer context window is clearly beneficial, and setting different context windows for different chunks can further improve the performance. This confirms our conjecture that contextual information is important for representing speech signals, and exploiting multi-scale contextual information is especially important.

An interesting comparison is between the Speech-MLP-S model with window $\{3, 7, 9, 11\}$ and the Speech-MLP-L model with window $\{1\}$. The parameters of the two models are comparable, but the latter model does not involve any chunk splitting and context expansion. The clear advantage of the Speech-MLP-S model demonstrated that the performance improvement with larger and multi-scale context windows (ref. performance of Speech-MLP-S or Speech-MLP-L with different windows) is due to the newly designed Split& Glue structure, rather than the increase in parameters. This in turn demonstrated the value of domain knowledge: if we can exploit it appropriately, it is possible to design very parsimonious models.

Table 3: Performance of speech-MLP on KWS tasks with different configurations.

| Models | Window | Size | Accuracy% | | | | |
|---|---|---|---|---|---|---|---|
| | | | V2-35 | LW-10 | LW-100 | LW-1000 | LW-10000 |
| Speech-MLP-S | $\{3,7,9,11\}$ | 180k | 97.14 | 95.03 | 90.91 | 90.16 | 89.16 |
| | $\{3,3,3,3\}$ | 138k | 96.74 | 94.83 | 90.25 | 89.20 | 87.89 |
| | $\{3\}$ | 108k | 96.46 | 94.31 | 89.11 | 87.81 | 85.69 |
| | $\{1\}$ | 88k | 87.67 | 66.06 | 36.96 | 21.87 | 22.03 |
| Speech-MLP-L | $\{3,7,9,11\}$ | 480k | 97.31 | 95.37 | 92.11 | 91.50 | 90.82 |
| | $\{3,3,3,3\}$ | 337K | 96.96 | 95.29 | 91.52 | 90.83 | 89.89 |
| | $\{3\}$ | 239k | 96.95 | 94.79 | 90.26 | 89.26 | 87.52 |
| | $\{1\}$ | 175k | 82.43 | 66.12 | 32.36 | 28.42 | 22.85 |
| Speech-MLP-XL | $\{3,7,9,11\}$ | 2375K | 97.60 | 95.80 | 93.22 | 93.27 | 93.01 |
| | $\{3,3,3,3\}$ | 1727K | 97.39 | 95.65 | 92.97 | 92.75 | 92.24 |
| | $\{3\}$ | 1291K | 97.50 | 95.45 | 92.44 | 92.16 | 91.55 |
| | $\{1\}$ | 1003K | 85.69 | 61.03 | 45.88 | 28.59 | 22.51 |

## 4.2 SPEECH ENHANCEMENT

Speech enhancement, which aims at inferring clean speech from its corrupted version (Benesty et al., 2006; Loizou, 2007; Das et al., 2020), is another fundamental task used to evaluate our model. We choose the Voicebank+Demand datasetValentini-Botinhao et al. (2016) to perform the SE test. It contains clean speech signals from the Voicebank dataset, includes 28 speakers for training and 2 speakers for testing. Noise signals of 40 types from the DEMAND atabase Thiemann et al. (2013) were selected and were mixed into the clean speech. After the mixing, the training set and testing set involve 11,572 and 824 clips respectively. We split the training utterances into segments of 3 seconds without overlap. This resulted into 17,989 training samples, each sampling consisting of a noise corrupted segment and the corresponding clean segment. The goal of SE is to learn a mapping function that converts a noisy segment to a clean segment.

### 4.2.1 SETTINGS

The architecture of our SE model is shown in Table 1. As input, the model receives a 257-dimensional log-magnitude spectrum. The extracted features are first projected by a linear layer and reduced to 256-dimensional feature vector, which are then forwarded to 10 stacked speech-MLP blocks. The output from the last speech-MLP block is re-projected to 257-dimensional feature vector. After a hard-sigmoid function Courbariaux et al. (2015), the value of the output units correspond to the ratio masks on the 257-dimensional input log-magnitude spectrum. The clean speech signal is estimated by applying the ratio masks onto the noisy spectrum and reusing the noisy phase.

Table 4: SE results on VoiceBank+Demand dataset.*The size of T-GSA was estimated from the structure reported in the original paper, which involves 10 transformer layers, and the dimension of the input/output is 1024.

| Models | PESQ | CSIG | CBAK | COVL | Size |
|---|---|---|---|---|---|
| Unprocessed speech | 1.97 | 3.37 | 2.49 | 2.66 | - |
| SEGAN (Pascual et al., 2017) | 2.16 | 3.48 | 2.94 | 2.8 | - |
| TF-GAN (Soni et al., 2018) | 2.53 | 3.8 | 3.12 | 3.14 | - |
| MDPHD (Kim et al., 2018) | 2.7 | 3.85 | 3.39 | 3.27 | - |
| Metric-GAN (Fu et al., 2019) | 2.86 | 3.99 | 3.39 | 3.42 | - |
| Metric-GAN+ (Fu et al., 2021a) | 3.15 | 4.14 | 3.16 | 3.64 | - |
| Metric-GANU (Fu et al., 2021b) | 2.45 | 3.47 | 2.63 | 2.91 | - |
| PHASEN (Yin et al., 2020) | 2.99 | 4.21 | 3.55 | 3.62 | - |
| T-GSA (Kim et al., 2020) | 3.06 | 4.18 | **3.59** | 3.62 | 60M* |
| Speech-MLP (Ours) | **3.08** | **4.28** | 3.50 | **3.70** | 636K |

More details of the settings can be found in Appendix. The performance of the proposed model is compared to six benchmarks. Note that we focus on models trained without extra data, or extra models for knowledge distillation. The reader can find details on these enhancement methods in the references presented in Table 4. Following the convention on this test set, we report the results of four metrics: PESQ, BAK, SIG and OVL (Hu & Loizou, 2007).

### 4.2.2 RESULTS

The results are shown in Table 4, where we choose 6 baseline systems for comparison. Among these systems, T-GAS (Kim et al., 2020) is based on a transformer model. Similar to speech-MLP, the authors of T-GAS also noticed the importance of local context and designed an annealing approach to encourage attention on neighbour frames. However the attention is still global in nature, and the improvement with T-GAS was still attributed to the capacity of transformers in learning (not so) long-range dependency. Note that the size of the T-GAS model was not reported in the original paper, so we made an estimation according to the structure description.

The results shown in Table 4 demonstrated that our speech-MLP model outperformed all the six baselines. In particular, without modeling any long-range dependency, it outperformed T-GSA by almost 100 times smaller of model size. This comparative results challenge the assumption that the better performance of T-GSA over other baselines is due to its capacity of capturing long-range dependence in speech. Moreover, the model size of Speech-MLP is much smaller than T-GSA, and due to the concise architecture, the training is simple and fast. It provides a strong support for our argument that complex models are not necessarily the best, and a knowledge-based model may easily beat complex models with parsimonious parameters.

## 5 CONCLUSIONS

In this paper, we propose the speech-MLP model, a simple MLP architecture for speech processing tasks. Our main motivation was to find a compact solution that eliminates unnecessary complexity while being able to capture essential information from speech signals. By utilizing domain knowledge of speech, we designed a simple yet effective structure that involves only linear transform and normalization. The main ingredient is a split & glue structure, which splits input features into multiple chunks and makes them accounting for different contexts. This knowledge-based design reflects several properties of speech signals, including temporal variance, frequency symmetry, and short-term dependency. The experimental results on keyword spotting and speech enhancement demonstrated that speech-MLP is highly effective: with much less parameters and computation, it can beat larger and more elaborately designed models including transformers.

Much work remains, for example, how to design a better chunking and context; how to make the model even smaller (e.g., removing unnecessary residual connections); how to trade off the complexity in chunks and in depth. The ultimate goal is to design a light-weighted, sufficiently powerful and generalizable component for speech feature extraction. We believe the knowledge-driven feature extractor benefits general speech processing tasks, such as speech recognition and understanding.

## 6  REPRODUCIBILITY STATEMENT

We made the following efforts to ensure that the results reported in the paper can be reproduced by other researchers.

- We will release the code on github, so everyone can download
- The datasets used in this paper are all publicly available for researchers
- We documented the required python environment and provided a step-by-step guidance for the reproduction
- We fixed the random seed in the code, so that others can reproduce our result exactly.

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

# A APPENDIX A: DETAILS OF KWS EXPERIMENT

In this section, we present the details of the KWS experiment. We start with the system architecture, followed by the data preparation. We then present the training methods and the hyperparameters used in the experiments.

Table 5: Experimental settings on keyword spotting

|  | Parameter | Value |
| --- | --- | --- |
|  | win_len | 512 |
|  | win_hop | 160 |
| Feature Extraction | fft_len | 480 |
|  | n_mel | 80 |
|  | n_mfcc | 40 |
|  | window | hann |
|  | Time masks | 2 |
| Spec Aug | Time mask bin | [0, 15] |
|  | Frequency Masks | 2 |
|  | Frequency mask bin | [0, 7] |
|  | init lr | 1.0e-03(V2-35)/1.0e-02(LW) |
|  | label smoothing | 0.1 |
|  | Schedule | cosine |
| Learning Parameters | end lr | 1.0e-05(V2-35)/1.0e-04(LW) |
|  | Optimizer | AdamW |
|  | Weight Decay | 1.0e-4 |
|  | Dropout | 0.1 |

## A.1 SYSTEM ARCHITECTURE

Prior to feature extraction, speech signals are resampled to 16 kHz if needed. Then, we use librosa[4] to extract a 40-dimensional MFCC features. The parameters used to extract these features are presented in Table 5. Global mean and variance is also applied to normalize the extracted features. These statistics are calculated using the respective training set of each task. After that, the features are fed into the model shown in Figure 3.

Specifically, a linear transformation (Linear 0) operates on the normalized MFCC features, projecting them to 128-dimensional embeddings. These embeddings are then forwarded to stacked Speech-MLP blocks (4 blocks in our KWS study) to extract multiscale contextual representations. For each speech utterance, the last Speech-MLP block outputs a sequence of context-rich representations, and then a max pooling operation is adopted to aggregate this sequence to a single utterance-level representation. This representation is then passed to a $128 \times 128$ linear transformation and a GELU nonlinear activation function. It is then further processed by a $128 \times M$ linear transformation and a softmax nonlinear activation, where $M$ is the number of keywords. The final output of the above process is a vector that represents the posterior probabilities that the original speech utterance belongs to each keyword.

## A.2 DATA PREPARATION

### A.2.1 GOOGLE SPEECH COMMANDS

The google speech commands **V2-35** contains 35 classes. The data can be obtained at the provider's website[5]. There are $84,843$ training samples in total, with strictly no overlapping between training, validation and test sets.

---

[4] https://librosa.org/doc/latest/index.html
[5] http://download.tensorflow.org/data/speech_commands_v0.02.tar.gz

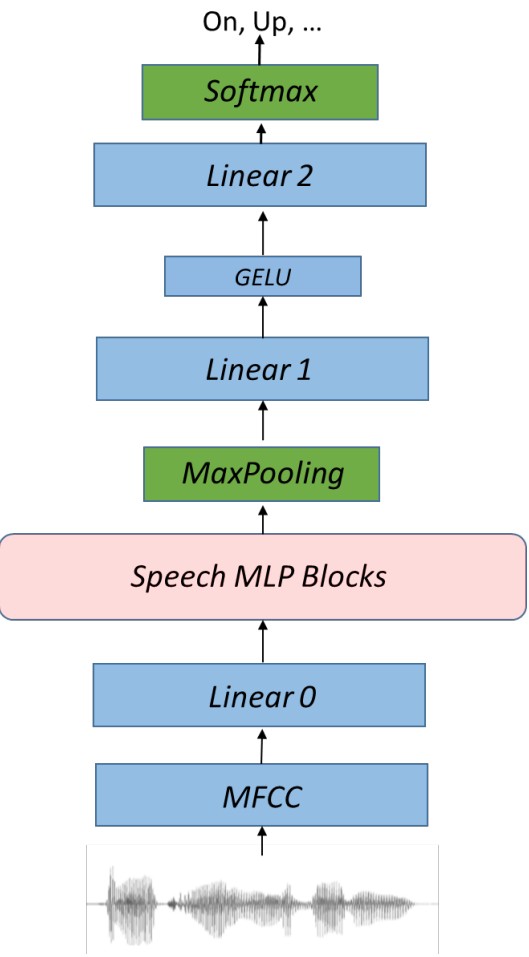

Figure 3: The system architecture used in the keyword spotting experiment.

Data augmentation techniques have been used to increase the training data by 9 times. Combined with the original data, we have $848,430$ training samples in total. We fixed the random seed to be $59185$ when producing the augmented samples. Following are the augmentation strategies adopted in this work:

- Noise perturbation: the noise perturbation script provided by the organizer of the DNS challenge is used to add background noise to clean speech[6]. The SNR factor is randomly sampled from $[5, 10, 15]$ with equal probabilities;

- Time shifting: time shifting is applied in the time domain. It shifts the waveform by a time-shift factor $t$ sampled from $[-T, T]$. In our experiments we set $T = 100$. When $t < 0$, the waveform is shifted left by $t$ samples and $t$ zeros are padded to the right side. When $t > 0$, the waveform is shifted right by $t$ samples and $t$ zeros are padded to left side;

- Resampling: the resample function from scipy (scipy.signal.resample) is used to perform resampling augmentation, which changes the sampling rate slightly. Specifically, given a parameter $R$, a resampling factor $r$ is drawn from $[1 - R, 1 + R]$, and the augmented sample is obtained by changing the sampling rate to $r \times 16000$. $R$ is set to $0.15$ in our experiments.

---

[6]We use the *segmental_snr_mixer* function from `https://github.com/microsoft/DNS-Challenge/blob/master/audiolib.py`

After the above augmentation, the original speech and the augmented speech are further corrupted by SpecAug (Park et al., 2019). The setting of SpecAug is shown in Table 5. Note that SpecAug does not enlarge the dataset.

### A.2.2 LIBRIWORDS

The LirbriWords dataset is a larger and more complex dataset. The samples are extracted from the json files provided by the providers[7]. We follow the task definition of the dataset provider, and the details are given below.

- LibriWords 10 (**LW-10**): this task contains 10 keywords, including "the", "and", "of", "to", "a", "in", "he", "I", "that", and "was". There are $1,750$k samples in total, and they are spit to a training set ($1,400$k), a validation set ($262,512$) and a test set ($87,501$).

- LibriWords 100 (**LW-100**): a more challenging task that contains 100 keywords. There are $1,512$k training samples, $189,010$ validation samples and $188,968$ samples, totalling $1,890$k samples.

- LibriWords 1000 (**LW-1K**): with increased difficult, this task contains 1000 keywords. The training set involves $2,178k$ samples, and the validation set and the test set contain $272,329$ samples and $271,858$ samples respectively.

- LibriWords 10000 (**LW-10K**): The most challenging task presents 9998 keywords. The training set contains $2,719k$ training samples, $339,849$ validation samples and $335,046$ test samples.

Given the large number of samples, data augmentation was not required for this task. We only performed SpecAug (Park et al., 2019) based on the settings presented in Table 5.

### A.3 TRAINING PARAMETERS

The parameters used during training are specified in Table 5. Further details are presented below.

- the cross entropy between the model prediction and the ground truth is used as loss function;

- The optimizer used in all the experiments is AdamW. The initial learning rate is set to 0.01, and cosine annealing is applied to adjust the learning rate from 0.01 to 0.0001;

- Dropout is applied onto the residual connections within the speech-MLP block, with the dropout rate set to 0.1;

- Label smoothing is employed to prevent the over-confidence problem. The smoothing factor is set to 0.1;

- In the **V2-35** experiment, the models are trained for 100 epochs and 10 epochs warmup is applied, In the **LibriWords** experiment, the models are trained for 20 epochs without warmup;

- In both the experiments, the model of each epoch is evaluated on the evaluation set, and the checkpoint that performs the best on the validation set is saved to report the performance on test set;

- We fix the random seed to be 123 in all the ablation study experiments, for the sake of reproducibility.

## B APPENDIX B: DETAILS OF SE EXPERIMENT

### B.1 SYSTEM ARCHITECTURE

The model architecture has been presented in Figure 4. The primary goal is to learn a mapping function that converts noisy magnitude spectrum to clean magnitude spectrum. The model output

---

[7]https://github.com/roman-vygon/triplet_loss_kws

Table 6: Experimental settings on speech enhancement

|  | Parameter | Value |
|---|---|---|
|  | win_len | 512 |
|  | win_len | 480 |
| Feature Extraction | win_hop | 160 |
|  | fft_len | 512 |
|  | window | hann |
|  | init lr | 1.0e-02 |
|  | Schedule | cosine |
|  | T | 3000 |
| Learning Parameters | end lr | 1.0e-04 |
|  | Warmup | 30 |
|  | Optimizer | AdamW |
|  | Epoch | 1000 |

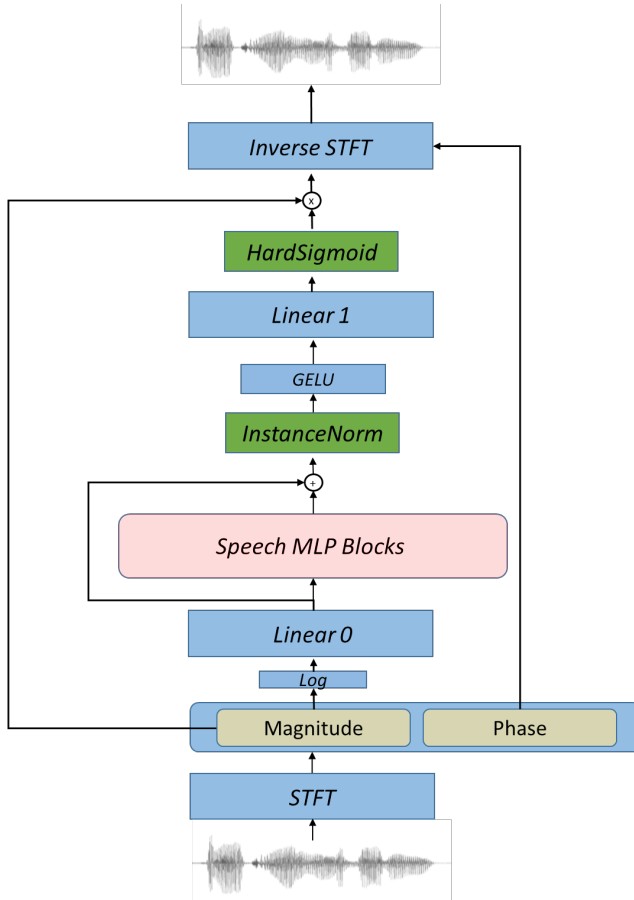

Figure 4: The system architecture used in the speech enhancement experiment.

predicts the soft ratio masks, that can be applied to the noisy magnitude spectrum to estimate the mangitude spectrum of the clean speech. Combining the denoised magnitude spectrum and the phase spectrum of the original noisy speech, one can attain the denoised waveform by inverse STFT.[8]

---

[8]We used the STFT class implemented in the torch-mfcc toolkit(`https://github.com/echocatzh/torch-mfcc`).

More specifically, 257-dimensional log-magnitude spectrum is firstly extracted from the noisy speech as the acoustic features, following the configuration shown in Table 6. Then a linear layer follows and transfers the input features to 256-dimensional vectors $PreX$. The transformed feature vectors are then forwarded to 10 Speech-MLP blocks, and the output from the last block, denoted by $PostX$, involves multiscale contextual information. Afterwards, a residual connection adds $PreX$ and $PostX$ together, and instance normalization is applied to regulate temporal variance. Finally, another linear transform and a non-linear HardSigmoid activation projects the normalized feature to a masking space where the dimensionality is the same as the input feature, corresponding the ratio mask $M \in [0, 1]$ on the noisy magnitude spectrum.

### B.2 Loss Functions

The loss function of our model is computed based on the discrepancy between the denoised speech $X_d$ and the clean speech $X_c$. The entire loss consists of two parts: (1) the distance on power-composed magnitude spectrum, denoted by $L_{mag}$, and (2) the distance on power-compressed STFT, denoted by $L_{stft}$. We use a single frame to demonstrate this computation, where the real loss should compute the average of $L$ on all the frames.

$$D_{real}, D_{imag} = STFT(X_d)$$
$$C_{real}, C_{imag} = STFT(X_c)$$
$$D_{mag} = \sqrt{D_{real}^2 + D_{imag}^2}$$
$$C_{mag} = \sqrt{C_{real}^2 + C_{imag}^2}$$
$$L_{mag} = (C_{mag}^{0.3} - C_{mag}^{0.3})^2$$
$$D_{real}^{0.3} = \frac{D_{mag}^{0.3}}{D_{mag}} \times D_{real}$$
$$D_{imag}^{0.3} = \frac{D_{mag}^{0.3}}{D_{mag}} \times D_{imag}$$
$$C_{real}^{0.3} = \frac{C_{mag}^{0.3}}{C_{mag}} \times C_{real}$$
$$C_{imag}^{0.3} = \frac{C_{mag}^{0.3}}{C_{mag}} \times C_{imag}$$
$$L_{stft} = \left\{ (C_{real}^{0.3} - D_{real}^{0.3})^2 + (C_{imag}^{0.3} - D_{imag}^{0.3})^2 \right\}^2$$
$$\mathcal{L} = 10 \times L_{mag} + L_{stft}$$

### B.3 Training parameters

The parameters for model training are summarized in Table 6. Specifically, the model was trained for 1000 epochs using the adamw optimizer. The initial learning was set to 0.01, and a cosine annealling learning scheduler was used to adjust the learning rate from 0.01 to 0.0001 in 3000 steps. Warmup was applied and involved 30 epochs . The model was evaluated on the evaluation set every epoch, and the best checkpoint (in terms of PESQ) on the evaluation set was saved. The results are reported in terms of four metrics: PESQ, BAK, SIG and OVL (Hu & Loizou, 2007).[9]

## C Appendix C: Pseudo code for Split & Glue

---

[9]The evaluation script is pysepm (https://github.com/schmiph2/pysepm).

---

**Algorithm 1** Pseudo code for Split & Glue

---

**Input Sequence:** $X \in \mathcal{R}^{H \times T}$: sequence of acoustic features of $T$ frames and $H$ dimensions
**Input Parameter:** $w = \{w^0, w^1, ..., w^K\}$: window sizes of the $K$ chunks
**Input Parameter:** $p = \{p^0, p^1, ..., p^K\}$: padding definition for the $K$ chunks
**Input Parameter:** $s$: stride in context expansion
**Output** $Y \in \mathcal{R}^{H \times T}$: sequence of output features of $T$ frames and $H$ dimensions

**Ensure:** $H\%K = 0$
$\quad \{X^1, ..., X^K\} = chunk(X, H, K)$ $\qquad\qquad\qquad \triangleright$ Split $X$ to $K$ pieces on the channel dimension
$\quad$ **for** $k$ in range($K$) **do**
$\quad\quad X_w^k = unfold(X^k, w^k, p^k, s)$ $\qquad\qquad\qquad \triangleright$ Context expansion by unfolding
$\quad\quad Y^k = W_A^k X_w^k + b_A^k$ $\qquad \triangleright$ Linear projection A for each chunk, where $W_A^k = [\hat{H}, w^k \times H/K]$
$\quad$ **end for**
$\quad Y^G = [Y^0; Y^1, ..., Y^K]$ $\qquad\qquad\qquad\qquad \triangleright$ Concatenate $Y^k$ along channel dimension
$\quad Y^G = GELU(Y^G)$
$\quad Y = W_B Y + b_B$ $\qquad \triangleright$ Linear projection B to glue the chunks, where $W_B = [H, K \times \hat{H}]$

---

