# OpenReview forum: "Speech-MLP: a simple MLP architecture for speech processing"
_ICLR.cc/2022/Conference — ICLR 2022 Submitted_

### Official Review · Reviewer_Qsws · 2021-10-18

**Correctness:** 3
**Technical Novelty And Significance:** 2
**Empirical Novelty And Significance:** 2
**Recommendation:** 5
**Confidence:** 4

**Main Review:**

Strengths:
- The paper is well written and clearly describes the model used and the experiments performs. The descriptions are accompanied with high-quality diagrams and specifications for hyperparameters. The motivations behind model choices are also explained.

Weaknesses:
- One main contribution of the paper, the "split and glue" layer, seems like a groupwise convolution layer. A groupwise convolution layer can be implemented by unfold (im2col) followed by a linear layer and concatenation. An unfold followed by a linear layer *is* a convolution. This significantly reduces the novelty and undermines the claim that the "architecture involves simple linear transformations only". Prior art on keyword spotting with convolutional networks is not hard to find (e.g. [0]). If my analysis is incorrect, and "split-and-glue" *cannot* be reduced to a groupwise convolution or at least something very similar to it, the paper can be much improved by a comparison to groupwise convolution.
- The paper compares KWS results on Google Speech Commands and LibriWords. Although Google Speech Commands is a well known dataset for KWS, LibriWords is not commonly used and as far as I can tell has only ever been evaluated with once in a single paper that proposed this dataset. I would focus on Google Speech Command results, as that dataset has been widely studied and benchmarked against. Although extra datasets do not *reduce* the quality of the paper, evaluating on LibriWords does not add much to the paper.
- A key part of the claim is that speech-MLP is "SOTA" on KWS. "It can be observed that the speech-MLP-S and speech-MLP-L outperform all the benchmarks for all task". However, upon cursory examination of recent literature, this claim seems dubious. Matchboxnet ([1]) achieves 97.37% on V2-35 with 140k parameters. Keyword Transformer (KWT) [2] achieves 97.51%. Note that KWT is even *cited* and compared against, but the result from KWT that the authors choose to cite is KWT1 (which speech-MLP outperforms) and not KWT2 or KWT3 (which outperform speech-MLP). (Arguably, this is because KWT2 and KWT3 have way more parameters, but this is not clear at all in the paper, and it's unclear if this is important). More recent works (Audiomer and Audio Spectrogram Transformer [4]) *significantly* outperform speech-MLP with 99.74 and 98.1%. The paper would be much improved by citing all relevant results rather than cherry-picking the results that support the claim of "speech-MLP is SOTA" and omitting others.
- Much of the introduction and motivation is based around model aesthetic cleanliness, that is, around claims that one model is 'more complex' than another and that simpler models are preferable. However, model complexity is subjective and not well defined. While the authors of the paper believe that transformers are a complex model and that the prosed speech-MLP model is "simple", this may not match the expectations of others, for whom a single architecture commonly known and used across industry and research is simpler than a custom architecture with a variety of dataset-specific tweaks. It may be best to leave subjective judgments of complexity out and focus on other benefits of the proposed model, e.g. parameter efficiency, inference speed, etc.
- One reason that speech-MLP may make a good architecture is parameter size and efficiency. In comparison to KWT2, which achieves similar / better results, it is much smaller. Other KWS papers will make comparisons of inference time as the model scales; this may help make the case for this architecture as well.
- Unlike many of the cited papers, none of the values have standard deviations or confidence intervals. It would help understand the results to have these.
- May be good to compare to more recent VoiceBank-DEMAND papers as well.

[0] https://static.googleusercontent.com/media/research.google.com/en//pubs/archive/43969.pdf
[1] https://arxiv.org/pdf/2004.08531.pdf
[2] https://arxiv.org/pdf/2104.00769.pdf
[4] https://arxiv.org/pdf/2104.01778v3.pdf

**Summary Of The Paper:**

The paper proposes a new architecture for speech processing, dubbed "speech-MLP". Besides input layers and output task-specific layers, speech-MLP consists of linear layers, residual connections, instance or layer normalization, GELU activations, and a layer called the "split-and-glue" layer. The split-and-glue layer splits a [batch, time, channels] tensor into N chunks of shape [batch, time, channels / N] along the channels dimension, then applies an unfold operation to provide temporal context, then a linear layer (with a different matrix for each chunk), then concatenates the results back together. Speech-MLP is tested on keyword spotting (with 2 datasets: Google Speech Commands and LibriWords) as well as speech enhancement (voicebank + demand).

**Summary Of The Review:**

The paper is well written and easy to understand with clear diagrams. However, I have some significant concerns about the split-and-glue layer (which seems to me to be a groupwise or separable convolution) and the keyword spotting comparisons (which omit results which outperform speech-MLP, then claim that speech-MLP surpasses the SOTA results). With these (and other more minor) concerns, it is hard to evaluate the novelty and impact of the paper, but these issues would be fixed with improved writing, literature review, and hyperparameter search / comparison.

---

> ### Author Response · Authors · 2021-11-16
> **We thank reviewer for taking the time to assess our submission. Bellow, we address all the relevant comments provided by the reviewer.**
>
> * One main contribution of the paper, the "split and glue" layer, seems like a groupwise convolution layer. A groupwise convolution layer can be implemented by unfold (im2col) followed by a linear layer and concatenation. An unfold followed by a linear layer is a convolution. This significantly reduces the novelty and undermines the claim that the "architecture involves simple linear transformations only". Prior art on keyword spotting with convolutional networks is not hard to find (e.g. [0]). If my analysis is incorrect, and "split-and-glue" cannot be reduced to a groupwise convolution or at least something very similar to it, the paper can be much improved by a comparison to groupwise convolution.
> Response: We agree with the reviewer that split & glue is similar to group-wised convolution. The difference is that speech-MLP contains multi-scale convolution, which made the architecture more flexible, and the multi-scale setting showed to be useful in the experimental results. Even though, the reviewer is correct that speech-MLP is similar to group-wised CNN. We made this point clear in the revised version. However, our main research goal is not a 'new architecture', but a 'simple architecture'. It is easy to design a new model by adding more complexity and showing that it works on some benchmark tests, but designing simple yet well-performed models is more challenging, although it is more desirable considering its economic and potential generalizability. We hope to show in this paper that a simple model can still achieve good performance in the benchmark test while using fewer parameters and less computation.
>
> * 2. The paper compares KWS results on Google Speech Commands and LibriWords. Although Google Speech Commands is a well known dataset for KWS, LibriWords is not commonly used and as far as I can tell has only ever been evaluated with once in a single paper that proposed this dataset. I would focus on Google Speech Command results, as that dataset has been widely studied and benchmarked against. Although extra datasets do not reduce the quality of the paper, evaluating on LibriWords does not add much to the paper.
>
> Response: We show the results on LibriWrods in order to show the generalization capacity of speech-MLP. Note that the Google Speech Commands set has been extensively tested and the performance reported is something over-tuned: some tricks on activation, structure, training scheme may lead to fluctuation on the reported results, which may shadow the true value of a model. Avoiding this problem is the main purpose of our study. Therefore, we test the model on a new Libriwords dataset, by using the exact same architecture and training scheme, to test if it still works. Moreover, Libriwords contain much more keywords, so the results are more convincing -- it is hard for one to tune for 10k words, but for 35 words, it is possible.
>
> * Much of the introduction and motivation is based around model aesthetic cleanliness, that is, around claims that one model is 'more complex' than another and that simpler models are preferable. However, model complexity is subjective and not well defined. While the authors of the paper believe that transformers are a complex model and that the prosed speech-MLP model is "simple", this may not match the expectations of others, for whom a single architecture commonly known and used across industry and research is simpler than a custom architecture with a variety of dataset-specific tweaks. It may be best to leave subjective judgments of complexity out and focus on other benefits of the proposed model, e.g. parameter efficiency, inference speed, etc.
>
> Response: We totally agree. We defined 'simple' in terms of parameter size and flops in the revision.
>
> * One reason that speech-MLP may make a good architecture is parameter size and efficiency. In comparison to KWT2, which achieves similar / better results, it is much smaller. Other KWS papers will make comparisons of inference time as the model scales; this may help make the case for this architecture as well.
>
> Response: Thanks for the reviewer's suggestion. We constructed a larger model that is comparable to KWT2 and found speech-MLP achieved better performance, with fewer flops.
> * Unlike many of the cited papers, none of the values have standard deviations or confidence intervals. It would help understand the results to have these.
> Response: We want to maintain the reproducibility of our paper so we fixed all the random seeds to 123 in all of our experiments. Nevertheless, we respect the reviewer's concern and will report the variance in the ablation study.
> * May be good to compare to more recent VoiceBank-DEMAND papers as well.
> Response: We thank the reviewer for the suggestion. We have added two recent papers, namely Metric-GAN+ and Metric-GANU, to table 4. We found that Metric-GAN+ indeed reported a higher PESQ, but that is due to the PESQ-oriented training, thus on other metrics, it is worse than speech-MLP.

---

> > ### Author Response · Authors · 2021-11-16
> > **We deleted the SOTA statement to make our point more clear that we want to find a simple and effective model with less parameters and flops..**
> >
> > * A key part of the claim is that speech-MLP is "SOTA" on KWS. "It can be observed that the speech-MLP-S and speech-MLP-L outperform all the benchmarks for all task". However, upon cursory examination of recent literature, this claim seems dubious. Matchboxnet ([1]) achieves 97.37\% on V2-35 with 140k parameters. Keyword Transformer (KWT) [2] achieves 97.51\%. Note that KWT is even cited and compared against, but the result from KWT that the authors choose to cite is KWT1 (which speech-MLP outperforms) and not KWT2 or KWT3 (which outperform speech-MLP). (Arguably, this is because KWT2 and KWT3 have way more parameters, but this is not clear at all in the paper, and it's unclear if this is important). More recent works (Audiomer and Audio Spectrogram Transformer [4]) significantly outperform speech-MLP with 99.74 and 98.1\%. The paper would be much improved by citing all relevant results rather than cherry-picking the results that support the claim of "speech-MLP is SOTA" and omitting others.
> >
> > Response:
> > (1) Firstly, we agree that 'speech-MLP achieves SOTA performance' is over strong. Our focus is on small models only. Considering the scope limitation, saying that the model is SOTA is not appropriate. We deleted the statement.
> > (2) The reason we choose to report the results of KWT-1 is that we are mostly interested in small models, and we want to compare our model with a similar scale in parameters. To respect the reviewer's comments, we tested a larger setting, named speech-MLP-XL in the paper and we compared it to KWT-2. The results showed that speech-MLP-XL has fewer parameters compared to KWT-2 but got better performance  (97.60% vs. 97.53%), and faster inference speed (0.228 vs 0.469 in GFLOPS). Note that KWT-3, an even larger model, does not show any superiority.
> > (3) Regarding the Audiomer model, we have checked their code and found that they misused the data in the V2-35 experiment: the test set is in the training set. This can be easily justified by checking their data script.
> > (4) Regarding the Audio Spectrogram Transformer, the authors used a pre-trained ViT on imagenet dataset to initialize the model. This pre-training may lead to clear performance improvement, and so it is not directly comparable with models trained from scratch, as speech-MLP did.

---

> > ### Author Response · Authors · 2021-11-17
> > **Mean and Variance on Speech-MLP-S on V2-35 by different random seeds.**
> >
> > We tested the model on five random seeds, the preliminary results show that the randomness of the model is not significant (compared with KWT-1 which is 96.85 (±0.07)), we are testing more random seeds to consolidate the result and current results are presented as follow:
> >
> > |    Models    | Random Seed |   Accuracy   |
> > |:------------:|:-----------:|:------------:|
> > | Speech-MLP-S |   123   |     97.14    |
> > |          |   24   |     97.07    |
> > |          |   2424   |     97.17   |
> > |          |   242424  |     97.15   |
> > |          |   24242424  |     97.22   |
> > | Speech-MLP-S |   mean&std  | 97.15(±0.05) |

---

### Official Review · Reviewer_EFsM · 2021-10-28

**Correctness:** 3
**Technical Novelty And Significance:** 2
**Empirical Novelty And Significance:** Not applicable
**Recommendation:** 3
**Confidence:** 4

**Main Review:**

I begin with the experiments because that is the positive part, at least for keyword spotting.  The performance is good, and that is in itself interesting.  This is especially true for the smaller architecture, where there are far fewer parameters.  Although the evaluation on speech enhancement produces good objective results, the measures are hardly significant.  A 0.02 improvement in PESQ is not regarded as significant, and the others are derivatives of PESQ.  Modern measures such as STOI and frequency weighted segmental SNR are missing.  The enhancement results can only be taken as preliminary, suggesting that a subjective test is worthwhile.

I find the method part less persuasive, beginning with the claim that the contribution is threefold.  The combination of propose, test and demonstrate is *one* contribution.  Thereafter I have multiple difficulties:
The authors claim early in the manuscript that they do not expect Speech-MLP to work for speech recognition.  Given the speech recognition is *the* main application of machine learning in speech processing, Speech-MLP is not really a good name.  Pragmatically, the longer range dependencies required for speech recognition are likely to come from other architectures.
The authors introduce their architecture simply by describing it.  Lacking is a description of Mixer MLP and how the new architecture differs from it.  My reading of this section is that Speech-MLP *is* Mixer-MLP with some minor modifications (it is not clear what they are).  Further, where in the introductory material, the authors claim that their solution is based on domain knowledge, I do not see how the proposed architecture addresses this where Mixer MLP would not.
The descriptive comparison with transformers is selective; it is true that transformers discard this type of domain knowledge, but *every* other speech processing architecture does take it into account.  A case in point is the TDNN used in many solutions.
In general, The method section should be rewritten to explain how and why the proposal differs from Mixer-MLP, and to place it in the context of other common signal processing techniques that also take such structure into account.
All results should include significance tests.

**Summary Of The Paper:**

The paper presents Speech-MLP, an architecture based on Mixer MLP but specific for speech signals.  The Mixer MLP architecture is argued to be appropriate for the particular structure of speech.  The architecture is compared to others on keyword spotting and speech enhancement.  In each case, the architecture outperforms other competitive solutions, often with fewer parameters.

**Summary Of The Review:**

Whilst some results are persuasive, showing that the architecture can perform as well as much larger ones, the method lacks rigor and contains no clear advance over other the methods on which it is based.

---

> ### Author Response · Authors · 2021-11-16
> **We thank reviewer for taking the time to assess our submission. Bellow, we address all the relevant comments provided by the reviewer.**
>
> * I begin with the experiments because that is the positive part, at least for keyword spotting. The performance is good, and that is in itself interesting. This is especially true for the smaller architecture, where there are far fewer parameters. Although the evaluation on speech enhancement produces good objective results, the measures are hardly significant. A 0.02 improvement in PESQ is not regarded as significant, and the others are derivatives of PESQ. Modern measures such as STOI and frequency weighted segmental SNR are missing. The enhancement results can only be taken as preliminary, suggesting that a subjective test is worthwhile.
>
> Response: Thanks for the reviewer's comments.
>
> (1) Regarding the SE results, we choose the metrics reported in the leaderboard: https://paperswithcode.com/sota/speech-enhancement-on-demand. Using these metrics, we can perform direct comparisons with others.
>
> (2) Regarding the 0.02 improvement in terms of PESQ, although it seems not significant in value, it is achieved with a much smaller model (600k vs 60M). Moreover, our training objective does not optimize PESQ (as T-GSA does), so the improvement should not be regarded as trivial or random.
>
>
> * I find the method part less persuasive, beginning with the claim that the contribution is threefold. The combination of propose, test and demonstrate is one contribution. Thereafter I have multiple difficulties: The authors claim early in the manuscript that they do not expect Speech-MLP to work for speech recognition. Given the speech recognition is the main application of machine learning in speech processing, Speech-MLP is not really a good name. Pragmatically, the longer range dependencies required for speech recognition are likely to come from other architectures. The authors introduce their architecture simply by describing it. Lacking is a description of Mixer MLP and how the new architecture differs from it. My reading of this section is that Speech-MLP is Mixer-MLP with some minor modifications (it is not clear what they are). Further, where in the introductory material, the authors claim that their solution is based on domain knowledge, I do not see how the proposed architecture addresses this where Mixer MLP would not. The descriptive comparison with transformers is selective; it is true that transformers discard this type of domain knowledge, but every other speech processing architecture does take it into account. A case in point is the TDNN used in many solutions. In general, The method section should be rewritten to explain how and why the proposal differs from Mixer-MLP, and to place it in the context of other common signal processing techniques that also take such structure into account. All results should include significance tests.
>
> Response:
> Thanks for the reviewers' valuable comments. We try to make some explanations one by one.
>
> (1) About the claim that 'we do not expect speech-MLP to work well for ASR, it is indeed misleading. It should be: speech-MLP is not very suitable for ASR by itself, but it can still be used as an encoder for ASR, and when combined with temporal models, ASR can be well conducted. In fact, our primary results on ASR show that speech-MLP indeed can achieve reasonable performance, comparable to Jasper, a full CNN ASR model.
> Therefore, our claim was not correct, and we had eliminated and clarified that speech-MLP is widely applicable as a speech encoder. For this reason, we keep the name speech-MLP, as it is general enough for various speech processing tasks.
>
> (2) About MLP-mixture, we didn't mention too much the difference because speech-MLP is very different from MLP-mixture: it has only 'channel mixing', but no 'patch mixing'. The relation is just that we were inspired by MLP-mixture to seek simple (small and fast) architectures. Anyway, we understand the reviewer's concern and gave more description about the difference between the two models in the revision.
>
> (3) We agree that many (or most) speech processing techniques implemented prior knowledge of speech signals, like TDNN. It is not our intention to argue speech-MLP is the only architecture that uses the prior. The main argument is that by considering the prior, the architecture can be designed parsimoniously, and the simplified architecture can still achieve good performance. We respect the reviewer's comments and give more discussion about the prior use in general speech processing techniques.
>
> (4) We will report the variance of the results in the ablation study, to show confidence. The significant test is not easy to conduct, as when compared to others, we mostly refer to the numbers published in the literature.

---

> > ### Author Response · Authors · 2021-11-17
> > **Mean and Variance on Speech-MLP-S on V2-35 by different random seeds.**
> >
> > We tested the model on five random seeds, the preliminary results show that the randomness of the model is not significant (compared with KWT-1 which is 96.85 (±0.07)), we are testing more random seeds to consolidate the result and current results are presented as follow:
> >
> > |    Models    | Random Seed |   Accuracy   |
> > |:------------:|:-----------:|:------------:|
> > | Speech-MLP-S |   123   |     97.14    |
> > |          |   24   |     97.07    |
> > |          |   2424   |     97.17   |
> > |          |   242424  |     97.15   |
> > |          |   24242424  |     97.22   |
> > | Speech-MLP-S |   mean&std  | 97.15(±0.05) |

---

### Official Review · Reviewer_cq7P · 2021-11-02

**Correctness:** 3
**Technical Novelty And Significance:** 2
**Empirical Novelty And Significance:** 2
**Recommendation:** 5
**Confidence:** 3

**Main Review:**

Strengths: The proposed model is relatively simple and easy to implement. It leverages some of the important characteristics of human speech such as temporal invariance, frequency asymmetry and short-term stationarity. The model is modular and can be easily integrated into any existing pipeline to process the features in a desired manner. Finally, it shows improvements on multiple tasks and across multiple datasets.


Weaknesses: This model appears to be convolution in disguise. The splitting operation is akin to breaking down the signal into multiple frequency bands and learning a convolution kernel for each band separately, i.e, separable convolutions. The unfolding operation or context expansion is where the model resembles the convolutional structure the most. It is similar to creating a circulant matrix and applying linear transformation to the expanded signal. The linear transformation operator is analogous to the convolutional kernel. Therefore, the authors, unknowingly, are proposing a convolutional neural network with residual connections and labelling it as 'split and glue' model. Further, convolutions can actively take advantage of temporal invariance or short-term stationarity of speech signal which seems to be one of the main drivers of performance gains in this model. Finally, the authors have not clarified if the data augmentation was done for the baseline models or not. The augmentation procedure can have a huge impact on the generalization of neural networks.

The experiments on keyword spotting use cross entropy loss for backpropagation. My suggestion to the authors would be to use the triplet loss for training as it outperforms the entropy based model in baseline comparisons. Additionally, the baseline methods in the speech enhancement task are mostly generative models which are perhaps not trained in a supervised setting. Therefore, comparison with some recently proposed supervised methods for speech enhancement would provide a good sense of the performance improvements. In the ablation study, it would be valuable for the speech and machine learning community to see how the proposed split and glue model performs without the outer residual connection in each block. While the paper compares the proposed approach to a convolutional model (Res-15), the presence of residual connections (inner+outer) and normalization (layer/instance) adds a lot of confounders in the analysis. Finally, a statistical test or error bars on the speech enhancement metrics will help validate the claims made by the authors.

**Summary Of The Paper:**

The authors have proposed a new general-purpose MLP architecture for speech processing and learning speech representation for tasks such as keyword spotting and speech enhancement. The proposed model processes the spectral representation of speech in multiple frequency/channel bands ('split' operations). The unfolding operation expands the context of each banded signal and applies linear transformations to it. These transformations are not shared across the chunks/bands which allows the model to learn chunk-wise relevant representations. Further, the unfolding procedure (context expansion) provides the benefit of learning segmental and supra-segmental properties of speech. The transformed outputs are concatenated together which is referred to as the 'glue' operation. Finally, the learned representations are passed into task-specific layers for generating output predictions. There are two residual connections in each split and glue block which ensures that there is a proper flow of gradients during the backpropagation to train the model.

The authors demonstrate the performance of the proposed mlp architecture on two problems, namely, keyword spotting, and speech enhancement. The model shows improvement over the state-of-the-art baseline techniques on multiple datasets (Google V2 and Libriwords) for keyword spotting. The ablation studies show how the choice of multiple linear operator kernels in 'split and glue' perform better than no split model. The speech enhancement experiment has been carried out on VoiceBank+Demand dataset and the authors show improvement by proposed technique across multiple metrics and baseline models. Therefore, the main contribution of this paper is to show that mlp architectures that are heuristically driven and are derived from domain knowledge can outperform state-of-the-art models for various prediction tasks.

**Summary Of The Review:**

The proposed architecture is similar to a convolutional neural network. The context expansion and chunk-wise processing is analogous to the concept of separable convolution. The experiments however, show the effectiveness of the model on keyword spotting and speech enhancement tasks. The proposed approach appears to take advantage of the structure of speech signals, specifically, temporal continuity. The authors have also conducted ablation studies to show how splitting the model into multiple frequency bands has desirable effects. It would be interesting to see how the 'split and glue' output performs on the relevant tasks without any residual connection. Further, comparison with supervised methods and error bars on the evaluation metric for speech enhancement will provide additional insight into the proposed technique.

---

> ### Author Response · Authors · 2021-11-16
> **We thank reviewer for taking the time to assess our submission. Bellow, we address all the relevant comments provided by the reviewer.**
>
> * This model appears to be convolution in disguise. The splitting operation is akin to breaking down the signal into multiple frequency bands and learning a convolution kernel for each band separately, i.e, separable convolutions. The unfolding operation or context expansion is where the model resembles the convolutional structure the most. It is similar to creating a circulant matrix and applying linear transformation to the expanded signal. The linear transformation operator is analogous to the convolutional kernel. Therefore, the authors, unknowingly, are proposing a convolutional neural network with residual connections and labelling it as 'split and glue' model. Further, convolutions can actively take advantage of temporal invariance or short-term stationarity of speech signal which seems to be one of the main drivers of performance gains in this model. Finally, the authors have not clarified if the data augmentation was done for the baseline models or not. The augmentation procedure can have a huge impact on the generalization of neural networks.
>
> Response:
> (1)  It is true that the split & glue operation is equal to group convolution (separable convolution with groups) with different kernel sizes if we regard the frame-wise operation as a 1-d temporal convolution. As we replied to the first reviewer, the reason that we did not call it convolution but 'speech-MLP' is that we want to follow the frame-wised perspective and focus on the feature (or channel) learning. From that perspective, the operation is more split & glue on the channel axis rather than convolution on the time axis. This frame-wised perspective is a long-standing convention in speech processing, and that is why the 1-d temporal convolution is often called TDNN in speech processing literature. We added this important point in the revision.
>
> (2) We clarified that data augmentation was used in all the baseline systems. This has been clarified in the revision.
>
> * The experiments on keyword spotting use cross entropy loss for backpropagation. My suggestion to the authors would be to use the triplet loss for training as it outperforms the entropy based model in baseline comparisons. Additionally, the baseline methods in the speech enhancement task are mostly generative models which are perhaps not trained in a supervised setting. Therefore, comparison with some recently proposed supervised methods for speech enhancement would provide a good sense of the performance improvements. In the ablation study, it would be valuable for the speech and machine learning community to see how the proposed split and glue model performs without the outer residual connection in each block. While the paper compares the proposed approach to a convolutional model (Res-15), the presence of residual connections (inner+outer) and normalization (layer/instance) adds a lot of confounders in the analysis. Finally, a statistical test or error bars on the speech enhancement metrics will help validate the claims made by the authors.
>
> Response:
>
> (1) About the triplet loss: thanks for the suggestion, and from the resnet results, it is true that triple loss is more effective. In this paper, the main comparative system is KWT, the transformer architecture. That is why we use the basic CE loss. But yes, the reviews' suggestion is helpful and we will try triple loss for speech-MLP training.
>
> (2) In the speech enhancement experiment, T-GSA is the main comparative model, and it should be regarded as 'supervised' if we notice that it optimize PESQ directly. We appreciate the reviewers' suggestion on comparison with more baselines, we also tried to do so. However, it seems not an easy task, as different models have their own design purpose. For our work, the main purpose is to demonstrate that a simple model like speech-MLP can achieve comparable performance as a more complicated model such as a Transformer. For this purpose, we mostly followed the test principle of T-GSA.
>
> (3) We didn't provide the mean + variance results because a particular goal of the research is to let the readers can fully reproduce our results reported in the paper, so we fixed the random seed. However, we respect the reviewer's suggestion and will present the variance of the Speech-MLP-S on V2-35 results in the ablation study to show the impact of different random seeds.

---

> > ### Author Response · Authors · 2021-11-17
> > **Mean and Variance on Speech-MLP-S on V2-35 by different random seeds.**
> >
> > We tested the model on five random seeds, the preliminary results show that the randomness of the model is not significant (compared with KWT-1 which is 96.85 (±0.07)), we are testing more random seeds to consolidate the result and current results are presented as follow:
> >
> > |    Models    | Random Seed |   Accuracy   |
> > |:------------:|:-----------:|:------------:|
> > | Speech-MLP-S |   123   |     97.14    |
> > |          |   24   |     97.07    |
> > |          |   2424   |     97.17   |
> > |          |   242424  |     97.15   |
> > |          |   24242424  |     97.22   |
> > | Speech-MLP-S |   mean&std  | 97.15(±0.05) |

---

### Official Review · Reviewer_zWqe · 2021-11-02

**Correctness:** 4
**Technical Novelty And Significance:** 3
**Empirical Novelty And Significance:** 3
**Recommendation:** 8
**Confidence:** 4

**Main Review:**

The paper is very well written, it is clearly motivated and presents impressive results with such a simple architecture. If the code becomes publicly available as the authors promised it will be very useful for the research community.

The ablation study is useful but it would be more complete if additional parameters were investigated, e.g., number of blocks. In Table 1 they are set to 4 and 10 somehow arbitrarily. In Table 3, more window sizes can be considered, e.g., 7 and 9. It is also not clear how the number of chunks impacts performance in Table 3.

Why not considering some intermediate architectures between small and large? It would be useful to report resorts for a M(edium) architecture as well.

Also have the authors considered other ways to create the chunks, e.g., across the time dimension?

Some typos
Appendix B.3 Specificaaly
A.2.2 closing quotation marks are used as opening quotation marks


**Summary Of The Paper:**

The paper proposes a simple architecture based on the multi-layer perceptron for extracting information from speech signals. The architecture is based on a new layer called split and glue and can capture multi-scale local temporal dependecies. It is evaluated on two different problems, keyword spotting and speech enhancement and achieves state-of-the-art performance.

**Summary Of The Review:**

Overall, this is an interesting study, the paper is clearly written and motivated and presents a novel architecture which leads to state-of-the-art results for keyword spotting and speech enhancement.

---

> ### Author Response · Authors · 2021-11-16
> **We thank reviewer for taking the time to assess our submission. Bellow, we address all the relevant comments provided by the reviewer.**
>
> * The ablation study is useful but it would be more complete if additional parameters were investigated, e.g., number of blocks. In Table 1 they are set to 4 and 10 somehow arbitrarily. In Table 3, more window sizes can be considered, e.g. 7 and 9. It is also not clear how the number of chunks impacts performance in Table 3.
>
> Response: Thanks for the valuable suggestion. The present setting is mostly designed for a fair comparison with other models (e.g, with similar model sizes). We also added more experiments on Table 3; other experiments suggested by the reviewer are ongoing, sorry for that.
>
> * Why not considering some intermediate architectures between small and large? It would be useful to report resorts for an M(edium) architecture as well.
>
> Response: The main design for the architecture is to have a reasonable comparison with existing models, for which we need some trade-offs. Anyway,
> we have reported the results with an XL model called speech-MLP-XL, to show the performance of models with different sizes. On all these models, the performance observed is consistent when compared with other models.
>
> * Also have the authors considered other ways to create the chunks, e.g., across the time dimension?
>
> Response: Thanks for the suggestion. Yes, it is possible to create chunks in time, which will further reduce the model size.
> We have mentioned this point as future work and will make more experiments.
>
> * Some typos Appendix B.3 Specificaly A.2.2 closing quotation marks are used as opening quotation marks
>
> Response: Thanks for the careful reading. The typos are fixed.

---

> > ### Comment · Reviewer_zWqe · 2021-11-18
> > **Comment**
> >
> > Thank you for your response. The need to use similar model sizes as other works is very clear, but on top of that additional ablation experiments can be conducted. It would very helpful if the results of these experiments are added to the revised manuscript.

---

### Official Review · Reviewer_b8HX · 2021-11-03

**Correctness:** 3
**Technical Novelty And Significance:** 3
**Empirical Novelty And Significance:** 3
**Recommendation:** 5
**Confidence:** 4

**Main Review:**

strengths
- novel neural network architecture by revisiting an MLP
- simple but effective architecture with fewer parameters than transformer.
- shows the effectiveness in two different speech processing tasks (especially the speech enhancement task only with 600K parameters looks very strong).

weaknesses
- the effectiveness of the Split & Glue layer is similar to the convolution operation with different kernel/stride sizes.
- the method cannot be applied to a sequence-to-sequence task like ASR (but the method can be used as an encoder of seq2seq tasks).
- although the performance is strong, the task is rather simple and limited. It does not attract a machine learning researcher in general.
- the paper needs more surveys

**Summary Of The Paper:**

This paper proposes an MLP-based neural network, which is designed for speech processing. The method is an alternative architecture to a transformer encoder and is applied to several speech processing tasks (command recognition and speech enhancement). The novel Split & Glue layer is used to capture multi-resolution speech characteristics. The method achieved state-of-the-art performance in both command recognition and speech enhancement tasks.

Other comments
- I don't think (Huang et al., 2020) is a representative work for transformer ASR. The following papers are more appropriate:
  - Dong, Linhao, Shuang Xu, and Bo Xu. "Speech-transformer: a no-recurrence sequence-to-sequence model for speech recognition." 2018 IEEE International Conference on Acoustics, Speech and Signal Processing (ICASSP). IEEE, 2018.
  - Karita, Shigeki, et al. "A comparative study on transformer vs rnn in speech applications." 2019 IEEE Automatic Speech Recognition and Understanding Workshop (ASRU). IEEE, 2019.
- It's better to mention how it is simple with quantitative measures in the abstract and introduction (model size, computational cost, or the number of code lines).
- the paper should refer conformer and discuss it. It becomes SOTA in many speech processing tasks now. Also, historically, many deep-learning-based speech processing methods were started from MLPs but the paper does not have enough surveys.
- please discuss the online/streaming capabilities. This is an important function for speech processing.




**Summary Of The Review:**

This paper proposes a novel neural network architecture based on MLP. The neural network architecture in speech processing becomes more complicated, and the proposed method can provide simple and alternative solutions. The effectiveness is also shown with command recognition, keyword search, and speech enhancement. My concern of this paper is that the application is rather simple. The method has a lot of potentials and may have more attention if they are applied to sequence-to-sequence tasks like ASR and TTS.

---

> ### Author Response · Authors · 2021-11-16
> **We agree with the most parts of reviewer's comments and modified our submission accordingly,  the architecture/experiments presented in this work are mostly speech-related, and our hope is that the lesson of design with prior can save a lot’ is helpful for ML researchers in general.**
>
> * I don't think (Huang et al., 2020) is a representative work for transformer ASR. The following papers are more appropriate:
> Dong, Linhao, Shuang Xu, and Bo Xu. "Speech-transformer: a no-recurrence sequence-to-sequence model for speech recognition." 2018 IEEE International Conference on Acoustics, Speech and Signal Processing (ICASSP). IEEE, 2018.
> Karita, Shigeki, et al. "A comparative study on transformer vs rnn in speech applications." 2019 IEEE Automatic Speech Recognition and Understanding Workshop (ASRU). IEEE, 2019.
>
> Response: We thank the reviewer for the suggested references. They have been added to the paper.
>
> * It's better to mention how it is simple with quantitative measures in the abstract and introduction (model size, computational cost, or the number of code lines).
>
> Response: We have clarified in the abstract and introduction that the proposed model is advanced in model size and GFLOPS.
>
> * the paper should refer conformer and discuss it. It becomes SOTA in many speech processing tasks now. Also, historically, many deep-learning-based speech processing methods were started from MLPs but the paper does not have enough surveys.
>
> Response: We added some comments/discussions in related work, to highlight the features of the proposed model when compared to conventional MLP. We also mentioned conformer and discuss its advantage.
>
> * Please discuss the online/streaming capabilities. This is an important function for speech processing.
>
> Response: We added some discussion. In fact, speech-MLP can perform online processing but with a buffer for a few frames in the future. According to the present config, this causes a 200 ms delay for Speech-MLP-(S/L). Reducing this looking-ahead buffer will allow a faster response.
>
>
> * The effectiveness of the Split \& Glue layer is similar to the convolution operation with different kernel/stride sizes.
>
> Response: We totally agree, thanks for the review (and other reviewers) to point
> it out, and we should have made this clearly stated. The split \& glue operation is equal to group convolution with different kernel sizes if we regard the frame-wise operation as a 1-d temporal convolution. We prefer the name `speech-MLP' rather than `speech-Conv' as we want to follow the frame-wised perspective and put the focus on the feature (or channel) learning. From that perspective, the operation is more split & glue on the channel axis rather than convolution on the time axis. In fact, this frame-wised perspective is a long-standing convention in speech processing, and that is why the 1-d temporal convolution is often called TDNN in speech processing literature. We added this important point in the revision.
>
>
> * the method cannot be applied to a sequence-to-sequence task like ASR (but the method can be used as an encoder of seq2seq tasks).
>
> Response: Yes, the proposed speech-MLP is an encoder, rather than a full end2end model. Therefore, it is fine to be used in ASR but only used as the encoder part (or, feature extraction). In fact, we have performed some experiments and found that Speech-MLP can achieve comparable performance as Jasper, a full CNN model. The combination of Speech-MLP and Transformer or RNN is under testing. The role of speech-MLP as an encoder (backbone) has been made clearly in the revised version (the conclusion section).
>
> * Although the performance is strong, the task is rather simple and limited. It does not attract a machine learning researcher in general.
>
> Response: We agree, the architecture/experiments presented in this work are mostly speech-related, and our hope is that the lesson of 'design with prior can save a lot is helpful for ML researchers in general.
>
>
> * the paper needs more surveys
>
> Response: We have extensively rewritten the related work section and linked the proposal to more existing work.

---

### Comment · Area_Chair_nVjK · 2021-11-15
**Please address reviewers' comments**

Dear Authors,

Please address the reviewers' comments. Thanks!

---

### Comment · Area_Chair_nVjK · 2021-11-24
**Please update your ratings if needed based on the authors' responses**

Dear Reviewers,

The authors have made detailed responses to all the reviews. Please take a look and see whether they address your concerns and update the ratings if necessary. Thanks for your help and expertise!

---

### Decision · Program_Chairs · 2022-01-20

**Decision:**

Reject

**Comment:**

This paper proposes an MLP-based neural network specifically designed for speech processing. The proposed Split & Glue layer is used to capture multi-resolution speech characteristics. The method achieved better performance in both command recognition and speech enhancement tasks.

Two major concerns raised by the reviewers:
The proposed split & glue layer is similar to convolution. Although the authors revised the paper with more clarification on the differences, the op is equivalent to frame-wise convolution which has been explored in speech literature. This limits the novelty of the paper.
The experimental justifications are relatively simple and limited. On the voice command and speech enhancement tasks presented in the paper, stronger and better baselines would be more convincing to justify the benefit of the proposed method. Moreover, testing on large scale ASR tasks instead of the relatively simple voice command task would be more convincing.

The decision is mainly based on the limited novelty and experimental justification.